# Long-term solid fuel use and risks of major eye diseases in China: A population-based cohort study of 486,532 adults

Ka Hung Chan[1,2], Mingshu Yan[1,3], Derrick A. Bennett[1,4], Yu Guo[5], Yiping Chen[1,3], Ling Yang[1,3], Jun Lv[6], Canqing Yu[6], Pei Pei[5], Yan Lu[7], Liming Li[6‡], Huaidong Du[1,3]*, Kin Bong Hubert Lam[1]*, Zhengming Chen[1,3‡], on behalf of the China Kadoorie Biobank Study group[¶]

1 Clinical Trial Service Unit and Epidemiological Studies Unit, Nuffield Department of Population Health, University of Oxford, United Kingdom, 2 Oxford British Heart Foundation Centre of Research Excellence, University of Oxford, United Kingdom, 3 MRC Population Health Research Unit, Nuffield Department of Population Health, University of Oxford, United Kingdom, 4 NIHR Oxford Biomedical Research Centre, Oxford University Hospitals NHS Foundation Trust, United Kingdom, 5 Chinese Academy of Medical Science, Beijing, China, 6 Department of Epidemiology and Biostatistics, School of Public Health, Peking University Health Science Center, Beijing, China, 7 NCD Prevention and Control Department, Suzhou Center for Disease Control and Prevention, Suzhou, China

‡ These authors are joint senior authors on this work.
¶ Membership of the China Kadoorie Biobank Study group is provided in the Acknowledgements.
* huaidong.du@ndph.ox.ac.uk (HD); hubert.lam@ndph.ox.ac.uk (KBHL)

## Abstract

### Background

Over 3.5 billion individuals worldwide are exposed to household air pollution from solid fuel use. There is limited evidence from cohort studies on associations of solid fuel use with risks of major eye diseases, which cause substantial disease and economic burden globally.

### Methods and findings

The China Kadoorie Biobank recruited 512,715 adults aged 30 to 79 years from 10 areas across China during 2004 to 2008. Cooking frequency and primary fuel types in the 3 most recent residences were assessed by a questionnaire. During median (IQR) 10.1 (9.2 to 11.1) years of follow-up, electronic linkages to national health insurance databases identified 4,877 incident conjunctiva disorders, 13,408 cataracts, 1,583 disorders of sclera, cornea, iris, and ciliary body (DSCIC), and 1,534 cases of glaucoma. Logistic regression yielded odds ratios (ORs) for each disease associated with long-term use of solid fuels (i.e., coal or wood) compared to clean fuels (i.e., gas or electricity) for cooking, with adjustment for age at baseline, birth cohort, sex, study area, education, occupation, alcohol intake, smoking, environmental tobacco smoke, cookstove ventilation, heating fuel exposure, body mass index, prevalent diabetes, self-reported general health, and length of recall period.

After excluding participants with missing or unreliable exposure data, 486,532 participants (mean baseline age 52.0 [SD 10.7] years; 59.1% women) were analysed. Overall, 71% of participants cooked regularly throughout the recall period, of whom 48% used solid fuels consistently. Compared with clean fuel users, solid fuel users had adjusted ORs of

stated research purposes. Details of how to access China Kadoorie Biobank data and details of the data release schedule are available from https://www.ckbiobank.org/site/Data+Access.

**Funding:** KHC is supported by the Nuffield Department of Population Health Early Career Research Fellowship and Oxford BHF Centre of Research Excellence (RE/18/3/24214). DAB is supported by the NIHR Oxford Biomedical Research Centre. DAB, YC, KBHL, ZC received grant from the UK Medical Research Council: Global Challenges Research Fund – Foundation Award (MR/P025080/1). ZC received grants from the Kadoorie Charitable Foundation in Hong Kong for the CKB baseline survey and the first re-survey. ZC received further grants from the UK Wellcome Trust (212946/Z/18/Z, 202922/Z/16/Z, 104085/Z/14/Z, 088158/Z/09/Z) for the long-term follow-up of CKB. JL and LL received grants from the National Key Research and Development Program of China (2016YFC0900500, 2016YFC0900501, 2016YFC0900504, 2016YFC1303904) and from the National Natural Science Foundation of China (91843302, 91846303, 81390540, 81390541, 81390544). The UK Medical Research Council (MC_UU_00017/1,MC_UU_12026/2 MC_U137686851), Cancer Research UK (C16077/A29186; C500/A16896) and the British Heart Foundation (CH/1996001/9454) provide core funding to the Clinical Trial Service Unit and Epidemiological Studies Unit at Oxford University for the project. The funders had no role in study design, data collection and analysis, decision to publish, or preparation of the manuscript.

**Competing interests:** The authors have declared that no competing interests exist.

**Abbreviations:** BMI, body mass index; CI, confidence interval; CKB, China Kadoorie Biobank; DALY, disability-adjusted life year; DSCIC, disorders of sclera, cornea, iris, and ciliary body; HR, hazard ratio; ICD-10, International Classification of Diseases, 10th revision; IOP, intraocular pressure; LMIC, low- and middle-income country; OR, odds ratio; SES, socioeconomic status.

1.32 (1.07 to 1.37, $p < 0.001$) for conjunctiva disorders, 1.17 (1.08 to 1.26, $p < 0.001$) for cataracts, 1.35 (1.10 to 1.66, $p = 0.0046$) for DSCIC, and 0.95 (0.76 to 1.18, $p = 0.62$) for glaucoma. Switching from solid to clean fuels was associated with smaller elevated risks (over long-term clean fuel users) than nonswitching, with adjusted ORs of 1.21 (1.07 to 1.37, $p < 0.001$), 1.05 (0.98 to 1.12, $p = 0.17$), and 1.21 (0.97 to 1.50, $p = 0.088$) for conjunctiva disorders, cataracts, and DSCIC, respectively. The adjusted ORs for the eye diseases were broadly similar in solid fuel users regardless of ventilation status. The main limitations of this study include the lack of baseline eye disease assessment, the use of self-reported cooking frequency and fuel types for exposure assessment, the risk of bias from delayed diagnosis (particularly for cataracts), and potential residual confounding from unmeasured factors (e.g., sunlight exposure).

## Conclusions

Among Chinese adults, long-term solid fuel use for cooking was associated with higher risks of not only conjunctiva disorders but also cataracts and other more severe eye diseases. Switching to clean fuels appeared to mitigate the risks, underscoring the global health importance of promoting universal access to clean fuels.

## Author summary

### Why was this study done?

- Household air pollution from solid fuel use has been linked to higher risks of cataracts and a range of acute eye symptoms, but most previous studies used relatively crude exposure assessment methods and cross-sectional or case–control designs and were relatively small.

- The relationships of long-term solid fuel use with common eye diseases other than cataracts, including conjunctiva disorders, keratitis, and glaucoma, are poorly understood.

### What did the researchers do and find?

- We analysed data from 486,532 adults aged 30 to 79 years recruited from 10 areas of China into the China Kadoorie Biobank during 2004 to 2008 to assess the associations of self-reported long-term solid fuel use for cooking with risks of conjunctivitis, cataracts, disorders of sclera, cornea, iris and ciliary body (DSCIC), and glaucoma during approximately 10-year follow-up.

- Multivariable logistic regression was used to estimate odds ratios comparing long-term clean fuels and solid fuels users, as well as those who had switched from solid to clean fuels prior to the initial baseline survey.

- Long-term solid fuel use was associated with 32%, 17%, and 35% higher risks of conjunctiva disorders, cataracts, and DSCIC, respectively, but not associated with glaucoma. Individuals who had switched from solid to clean fuels appeared to have smaller risks than those who used solid fuels persistently.

**What do these findings mean?**

- To our knowledge, this is one of the first cohort studies on the relationships between long-term solid fuel use and risks of multiple common eye diseases.

- Our findings support a significant association between solid fuel use and cataracts, but the strength of association appeared to be considerably weaker compared to that observed in previous studies; the associations with conjunctiva disorders and DSCIC indicate that solid fuel use may have more extensive harm on eye health, which should be further investigated.

## Introduction

Household air pollution from domestic solid fuels (e.g., coal and biomass) use is a leading risk factor of disease burden from cardiovascular and respiratory diseases [1]. Despite recent improvements, household air pollution still affects about a half of the world's population, including 452 million in China and 846 million in India [1]. Among the many human organs that could be affected by household air pollution, the eyes are exposed directly to high levels of fine particulate matter ($PM_{2.5}$) and carbon monoxide [2]. Not surprisingly, eye problems (e.g., eye pain, tearing, and redness) are some of the most commonly reported symptoms linked to household air pollution exposure [2–4]. Although these symptoms are temporary, prolonged exposure may result in major vision impairment or blindness, which could substantially undermine the productivity and quality of life of the sufferers and their families [5,6].

It has been estimated that globally household air pollution accounted for 14 million disability-adjusted life years (DALYs) through cataracts in women, making it the top modifiable risk factor of cataracts, which are the largest attributable cause of vision loss or impairment worldwide [5,7]. However, these estimates were mainly based on results from an early meta-analysis of 7 studies published before 2010, which were constrained by certain methodological limitations such as small sample size, use of cross-sectional or case–control designs, ambiguous exposure classification (based only on household fuel types), or unclear control selection methods (for case–control studies) [8]. More recently, a large cross-sectional study in India reported a substantially weaker association with cataracts, leaving great uncertainty as regards this relationship [9]. Evidence on the relationship of household air pollution with other major eye diseases, such as conjunctivitis or glaucoma, is even more limited, possibly due to difficulties in outcome ascertainment in low- and middle-income countries (LMICs) [4].

Using data from the China Kadoorie Biobank (CKB), we conducted one of the first cohort studies on long-term solid fuel use for cooking and risks of 4 major categories of eye diseases and the implications of switching from solid to clean fuels or having ventilated cookstoves on those risks.

## Methods

### Study design

Details of the study design and population characteristics of the CKB study have been published previously [10,11]. During 2004 to 2008, 512,715 adults aged 30 to 79 years and without any major physical or mental disabilities were recruited from 10 areas across China, selected

from the nationally representative Disease Surveillance Point system [12]. Trained health workers administered a computer-based questionnaire interview assessing socioeconomic, lifestyle, fuel use behaviour, and medical history, and conducted physical measurements (e.g., height, weight, and blood pressure) following standardised protocols. Periodic resurveys have been undertaken in a random subset (approximately 5%) of the cohort every 4 to 5 years, in order to collect repeated measurements and additional data for enhancement.

Ethical approvals were obtained from the Oxford University Tropical Research Ethics Committee, the Chinese Academy of Medical Sciences Ethical Review Committee, the Chinese Center for Disease Control and Prevention Ethical Review Committee, and the scientific review boards in each of the 10 regional centres. All participants provided written informed consent for participation and for access to their health records during follow-up. No formal statistical analysis plan was available for the present manuscript. A STROBE checklist for the report of observational cohort studies is included as a supporting information (see **S1 STROBE Checklist**).

## Assessment of fuel use behaviour and household air pollution exposure

The methods of assessing fuel use behaviour in CKB have been described in detail elsewhere [13,14]. Briefly, participants recalled, for up to their 3 most recent residences, the years of living (median [IQR]: 40 [29 to 50] years; ≥20 years in 91% of participants), their cooking frequency, primary cooking fuel (i.e., the fuel type used most frequently and for the longest duration), and the use of ventilated cookstoves (i.e., those with chimney or extractor). Individuals who cooked weekly or daily were classified as regular cooks, and their exposure status in each residence were defined according to the primary fuel type, with gas and electricity as clean fuels, and coal and wood as solid fuels. Data from the 3 residences were combined to classify individuals into different long-term fuel use categories ("always clean fuels," "solid to clean fuels," "always solid fuels," "never-regular cooks"). For secondary analysis, the "always solid fuels" category was separated according to duration of exposure (<20, 20 to 39, ≥40 years) and type of fuel used (always coal, mix of coal and wood, always wood). As described previously [15], duration of solid fuel use within the recall period was considered as an ordinal rather than a continuous metric, because participants only reported rounded number of years lived in their 3 most recent residences and we did not have complete fuel use history for all participants. Our previous analysis indicated a moderate consistency (weighted Kappa statistics approximately 0.65) in terms of self-reported fuel use between the baseline survey and resurvey [15].

Participants who switched from solid to clean fuels were further categorised according to the years since switching (with a median cutoff of 15 years), and their risks of developing selected eye diseases were compared to long-term clean fuels and solid fuels users. The availability of ventilated cookstove(s) across the reported residences was aggregated to approximate long-term solid fuels users' ventilation status ("never had ventilation," "always partial or complete ventilation," "always complete ventilation," "mixed"), whereby complete ventilation is defined as all cookstoves had a chimney or extractor hood.

## Follow-up and outcome definition

All participants were followed up through electronic linkages, using their unique personal identification number, to death and disease registries, and national health insurance databases that had almost universal coverage across 10 study areas [16]. This linkage method was designed to capture primarily disease events requiring treatment in hospitals or health insurance reimbursement. For a small proportion (approximately 5%) of individuals who died

outside of clinical settings, standardised verbal autopsy was conducted to ascertain the most probable cause of death [11]. During the follow-up period, 44,037 (8.6%) participants had died and 4,781 (0.9%) were lost to follow-up. Participants were censored upon death, loss to follow-up, or January 1, 2017, whichever came first. Disease events were coded according to the International Classification of Diseases, 10th revision (ICD-10), blinded to baseline information. This study examined the first events reported by January 1, 2017 for 4 major categories (i.e., with ≥1,000 cases recorded, to ensure reasonable statistical power) of eye disease, namely conjunctiva disorders (ICD-10: H10-H11), cataracts (H25, H26.9), disorders of sclera, cornea, iris, and ciliary body (DSCIC; H15-H22), and glaucoma (H40-H42) (see Table A in S1 Tables for further details).

## Statistical analysis

The present study excluded individuals who had missing data on body mass index (BMI; $n$ = 2); those who provided unreliable residential history (indicated by a difference between age and length of recall period >1 year; $n$ = 2,189), those who reported using unspecified fuels at any residence ($n$ = 3,342), and those who had ever switched from clean to solid fuels ($n$ = 15,150), because they are indicators of having potentially unreliable or unclear exposure profiles; and those who had cooked previously but stopped at baseline ($n$ = 19,669), because their decision to stop cooking may be related to the disease outcomes of interest, thus leading to reverse causation bias. After these exclusions, 486,532 participants remained in the main analyses. Although the group of never-regular cooks is not directly relevant to the research questions of interest, they are retained in the analyses for comparison.

Direct standardisation [17] was used to obtain age-, sex-, and study area-adjusted percentages or means of baseline characteristics to be compared across long-term cooking fuel exposure categories (see Supplementary methods in S1 Text for further details). Adjusted disease incidence rates were computed using the same approach. Delays in diagnosis or treatment are particularly common for slowly progressing, nonlethal eye diseases such as cataracts, especially in rural areas and among those at a lower socioeconomic status (SES) [18], who are more likely to be solid fuel users. Since conventional survival analysis examines time-to-event, the corresponding relative risk estimates would be more sensitive to biases that arise from the disproportionately longer delays in time-to-event among solid fuel users compared to clean fuel users (see Fig A in S1 Figs for further explanation). Therefore, the primary analyses employed logistic regression to estimate adjusted odds ratios (ORs) and 95% confidence intervals (CIs) for the first event of the outcomes of interest (as a binary endpoint, ignoring the time-to-event). Subsidiary analyses using Cox regression analysis with similar adjustments, yielding adjusted hazard ratios (HRs), were conducted for comparison. Confounders were identified based on existing evidence about the epidemiological linkages between the cooking fuel use, eye disease risk, and potential confounders, as described by VanderWeele [19]. In addition, several key covariates (e.g., study area, cookstove ventilation, and length of recall period) specific to the present study were included based on a priori knowledge about the confounding structure relevant to solid fuel use and a range of disease outcomes in our previous studies [13,15]. Subsequently, a standard stepwise modelling approach was employed to assess the statistical contribution of the potential confounders in improving the log-likelihood ratio statistics [19]. In the final models, we adjusted for age at baseline, birth cohort, sex, study area, education, occupation, alcohol intake, smoking, environmental tobacco smoke, cookstove ventilation, heating fuel exposure, BMI, prevalent diabetes, self-reported general health, and length of recall period (see Supplementary methods in S1 Text for more details).

Subgroup analyses and formal tests for multiplicative interaction (by fitting relevant interaction terms and undertaking likelihood ratio tests to assess the relevant $\chi^2$ and corresponding

*p*-values) by sex and smoking status were conducted to explore potential effect modification. Separate sensitivity analyses were conducted to assess risks of residual confounding and reverse causation bias through (1) additionally adjusting for diet, physical activity level (metabolic equivalent of tasks), and baseline random blood glucose level; (2) excluding participants who cooked weekly but not daily (*n* = 59,791); (3) excluding individuals whose recall period was <20 years (i.e., frequent-movers, *n* = 30,672); (4) excluding those with prevalent diabetes based on baseline screening and self-reported medical history (*n* = 28,298); (5) excluding those with poor self-reported health (*n* = 49,480); (6) excluding participants aged ≥65 years at baseline (*n* = 74,686); (7) excluding participants aged <40 years at baseline (*n* = 76,430); (8) excluding the first 3 years of follow-up; and (9) excluding cases diagnosed within 1 year after the diagnosis of another eye disease, respectively. The mutual associations between the outcomes investigated was assessed by logistic regression adjusting for age, sex, birth cohort, education, and occupation. Leave-one-out analysis was also conducted by excluding 1 of the 10 study areas at a time, to examine the sensitivity of the main results to regional variation of exposure and outcome patterns. In order to allow comparisons of the relative risk estimates of any 2 categories of exposure (not just with the reference group) in the tables and figures, the group-specific CIs of ORs (and HRs) were estimated using the variance of the log odds in each category as described previously [20]. This method has distinct advantages for studies with polychotomous risk factors and has been widely used in similar studies [21–25]. Conventional 95% CIs were reported when explicit comparisons between 2 groups were made. The present report focused on the point estimates and associated 95% CIs of ORs when describing the associations examined to avoid misinterpretation of *p*-values [26]. We used SAS version 9.3 for all analyses.

## Results

Of the 486,532 participants included, the mean (SD) baseline age was 52.0 (10.7) years; 59.1% were women; 73.1% reported cooking regularly, of whom 48.7% had always used solid fuels (defined as long-term solid fuel users), 26.9% had switched from solid to clean fuels, and 24.4% were long-term clean fuel users. Compared to long-term clean fuel users, long-term solid fuel users tend to be older, female, rural residents, less educated, agricultural workers, regular-smokers, exposed to passive smoking, and using solid fuels for heating (**Table 1**). They also had lower household income, were less likely to use ventilated cookstoves and to have prevalent diabetes, but more likely to report poor health status.

During a median (IQR) 10.1 (9.2 to 11.1) years of follow-up, there were 4,877 first events of conjunctiva disorders, 13,408 cataracts, 1,583 DSCIC, and 1,534 cases of glaucoma (**Table 2**). In general, the disease incidence rates tended to increase with age, although those aged ≥70 years had somewhat lower rates of conjunctiva disorders (91.9 versus 129.7 per 100,000 person-year) and DSCIC (32.3 versus 48.8) compared to those aged 60 to 69 years. The rates of DSCIC differed little between sexes, but the rates of other 3 eye diseases were higher in women than in men. The rates of conjunctiva disorders, cataracts, and DSCIC were higher in rural than urban residents, while the converse was true for glaucoma. The 4 endpoints were strongly related to each other, with adjusted ORs ranging from 3.46 (95% CI 3.12 to 3.84) between conjunctiva disorders and cataracts to 10.3 (7.68 to 13.8) between DSCIC and glaucoma (Table B in S1 Tables).

Compared with long-term clean fuel users, solid fuel users had higher risks of conjunctiva disorders (adjusted OR = 1.32, 95% CI 1.07 to 1.37), cataracts (1.17, 1.08 to 1.26), and DSCIC (1.35, 1.10 to 1.66), but not glaucoma (0.95, 0.76 to 1.18) (**Fig 1**). Those who had switched from solid to clean fuels had no apparent elevated risks of cataracts (1.05, 0.98 to 1.12) and

**Table 1. Baseline participant characteristics by long-term solid fuel use for cooking[1].**

| | Always clean | Solid to clean | Always solid | Never-regular cook[2] | Overall |
|---|---|---|---|---|---|
| Total participants, N | **86,821** | **95,722** | **173,288** | **130,701** | **486,532** |
| Age (years), mean (SD) | 47.5 (10.1) | 52.1 (10.0) | 54.7 (10.5) | 48.9 (11.0) | 52.0 (10.6) |
| Female, % | 49.8 | 83.5 | 77.4 | 11.7 | 59.1 |
| Urban, % | 86.9 | 82.8 | 9.8 | 42.9 | 43.4 |
| Middle school or above, % | 64.7 | 57.3 | 40.7 | 54.3 | 48.9 |
| Household income <20,000 Yuan/year, % | 18.9 | 20.1 | 39.0 | 24.2 | 28.5 |
| Occupation, % | | | | | |
| Agricultural worker | 23.5 | 26.4 | 48.2 | 36.1 | 42.4 |
| Factory worker | 14.9 | 15.0 | 13.1 | 15.9 | 14.0 |
| Office worker | 17.2 | 15.5 | 6.4 | 14.0 | 10.0 |
| Home-maker | 11.2 | 13.3 | 13.3 | 8.5 | 10.4 |
| Others[3] | 33.2 | 29.8 | 19.0 | 25.6 | 23.3 |
| Regular smoking in men, % | 64.8 | 64.0 | 68.0 | 68.5 | 67.6 |
| Regular smoking in women, % | 2.3 | 2.9 | 4.0 | 3.2 | 2.8 |
| Regular drinking in men, % | 41.0 | 44.7 | 37.4 | 37.6 | 38.0 |
| Regular drinking in women, % | 3.1 | 2.6 | 2.4 | 3.4 | 2.5 |
| Daily exposure to passive smoking, % | 40.0 | 42.4 | 41.7 | 42.3 | 41.5 |
| Long-term solid fuel use for heating, % | 26.5 | 24.4 | 48.8 | 35.8 | 36.6 |
| Presence of ventilated cookstoves, % | 80.1 | 80.0 | 69.8 | 76.5 | 76.3 |
| Body mass index (kg/m$^2$), mean (SD) | 23.9 (3.4) | 24.1 (3.4) | 23.4 (3.4) | 23.6 (3.2) | 23.6 (3.4) |
| Random blood glucose (mmol/L), mean (SD)[4] | 6.0 (2.3) | 6.1 (2.5) | 6.0 (2.3) | 6.2 (3.2) | 6.1 (2.3) |
| Prevalent diabetes, %[5] | 6.8 | 6.9 | 5.2 | 6.9 | 5.8 |
| Self-reported poor health, % | 8.8 | 9.3 | 11.8 | 11.4 | 10.2 |

[1]Means and percentages were adjusted for age, sex, and study area, where appropriate.

[2]Never-regular cook: individuals who reported cooking for monthly or less frequently throughout the recall period.

[3]Others: retiree, self-employed, unemployed, or undefined.

[4]Missing in 8,341 participants.

[5]Prevalent diabetes: self-reported prior diagnosis of diabetes or screen-detected diabetes based on baseline blood glucose level.

DSCIC (1.21, 0.97 to 1.50) and smaller elevated risks of conjunctiva disorders (1.21, 1.07 to 1.37). There was evidence of a multiplicative interaction between solid fuel use and smoking status and sex for cataracts, with the higher risk associated with solid fuel use restricted to women (1.11 [1.00 to 1.23] versus 1.05 [0.93 to 1.18] in men, $p_{Interaction} < 0.001$) or never-smokers (1.16 [1.05 to 1.28] versus 1.01 [0.88 to 1.15] in regular-smokers, $p_{Interaction} < 0.001$) only (**Figs 2 and 3**). No apparent evidence for a multiplicative interaction was found for other outcomes ($p_{interaction\_smoking}$ = 0.134 for conjunctiva disorders, 0.279 for DSCIC, 0.280 for glaucoma; corresponding $p_{interaction\_sex}$ = 0.054, 0.125, 0.067, respectively) (**Figs 2 and 3**).

Longer duration of solid fuel use appeared to be associated with higher risks of conjunctiva disorders, cataracts, and DSCIC (**Fig 4**). Among the long-term solid fuel users, there was little difference in the risks of conjunctiva disorders and cataracts by fuel types, while the higher risk of DSCIC appeared somewhat greater for long-term wood users (1.39, 1.12 to 1.71) than coal (1.22, 0.93 to 1.61) or mixed fuel (coal and wood) users (1.26, 0.94 to 1.70) (**Fig 5**).

In further analysis comparing long-term clean fuel users (fuel use duration median [IQR] duration = 35 [21 to 44] years) with individuals who had switched from solid to clean fuels, those who had switched for a longer duration (≥15 years) appeared to have even smaller elevated risks of conjunctiva disorders (1.15 [0.98 to 1.33] versus 1.28 [1.11 to 1.47]) and DSCIC

**Table 2. Distribution and rates (per 100,000 person-years) of eye disease examined according to age, sex, and study area.**

| Characteristics | N | Conjunctiva disorders | | | Cataracts | | | Disorders of sclera, cornea, iris, and ciliary body | | | Glaucoma | | |
|---|---|---|---|---|---|---|---|---|---|---|---|---|---|
| | | Event no. | Crude rate | Adjusted rate* | Event no. | Crude rate | Adjusted rate* | Event no. | Crude rate | Adjusted rate* | Event no. | Crude rate | Adjusted rate* |
| **Age, years (mean)** | | | | | | | | | | | | | |
| 30–39 (37.3) | 73,078 | 355 | 46.3 | 30.5 | 201 | 26.7 | 42.4 | 131 | 16.8 | 12.1 | 46 | 6.1 | 4.2 |
| 40–49 (44.8) | 145,560 | 1,231 | 81.6 | 90.1 | 978 | 66.1 | 104.9 | 431 | 28.5 | 26.4 | 209 | 14.3 | 14.1 |
| 50–59 (54.6) | 150,241 | 1,959 | 129.6 | 121.3 | 3,642 | 244.2 | 307.6 | 599 | 39.5 | 32.9 | 529 | 35.3 | 35.2 |
| 60–69 (64.7) | 86,981 | 1,086 | 129.7 | 143.7 | 5,922 | 735.6 | 666.8 | 329 | 39.6 | 48.8 | 556 | 67.4 | 67.6 |
| ≥70 (72.6) | 30,672 | 246 | 91.9 | 92.0 | 2,653 | 1041.2 | 685.4 | 93 | 35.7 | 32.3 | 194 | 75.4 | 73.5 |
| **Sex** | | | | | | | | | | | | | |
| Male | 199,032 | 1,724 | 87.1 | 86.8 | 4,839 | 251.7 | 231.4 | 663 | 33.7 | 33.4 | 474 | 24.8 | 23.5 |
| Female | 287,500 | 3,153 | 108.3 | 111.5 | 8,557 | 299.0 | 320.2 | 920 | 31.5 | 32.4 | 1,060 | 36.8 | 37.9 |
| **Study area** | | | | | | | | | | | | | |
| Rural | 275,178 | 3,984 | 145.0 | 147.4 | 7,602 | 279.0 | 305.9 | 1,368 | 49.3 | 50.0 | 728 | 26.5 | 27.8 |
| Urban | 211,354 | 893 | 42.5 | 41.6 | 5,794 | 281.2 | 252.0 | 215 | 10.9 | 9.9 | 806 | 39.0 | 36.1 |

*Rates were adjusted for age, sex, and study area, where appropriate.

(1.17 [0.89 to 1.53] versus 1.27 [1.00 to 1.61]) than those who had switched for <15 years (**Fig 6**). However, no such difference was observed for cataracts. In contrast to switching to clean fuels, the ORs for the eye disease outcomes among long-term solid fuel users were similar regardless of cookstove ventilation status (all *p*-values >0.05; Fig B in S1 Figs).

Sensitivity analyses with additional adjustment for confounders or exclusion of more individuals to further reduce exposure or outcome misclassification biases did not materially alter the results (Tables C and D in S1 Tables). Similarly, the leave-one-out analysis yielded consistent results (Table E in S1 Tables). The Cox regression analyses comparing long-term solid fuel users with clean fuel users yielded HRs of similar magnitude to the ORs generated in the primary analyses on conjunctiva disorders, DSCIC, and glaucoma, although the HR for cataracts was considerably smaller than the corresponding OR (1.06 [0.98 to 1.15] versus 1.17 [1.08 to 1.26]) (Table F in S1 Tables). Similar patterns were observed for Cox regression analyses on duration and types of solid fuel use (Tables G and H in S1 Tables).

## Discussion

In this large population-based cohort study of 486,532 Chinese adults, long-term use of solid fuels for cooking was associated with 17% to 37% higher risks of conjunctiva disorders, cataracts, and DSCIC. The elevated risks were somewhat greater in those exposed for a longer duration and somewhat smaller in those switching from solid to clean fuels but did not differ by specific types of solid fuels. In contrast, solid fuel use was not associated with the risk of glaucoma.

Most previous epidemiological studies on household air pollution and clinical eye diseases have primarily focused on age-related cataracts (i.e., among people >50 years of age, without known mechanical, chemical, or radiation trauma), the predominant type of cataracts in the general population. Notably, all these studies were relatively small, were unable to explore the temporality of association, and adopted ambiguous proxies (e.g., "cheap cooking fuel," "smoky cooking fuel," and household fuel/stove types) to define exposure or used inappropriately defined reference group (e.g., kerosene, "other types," and "non-users"). Their findings were

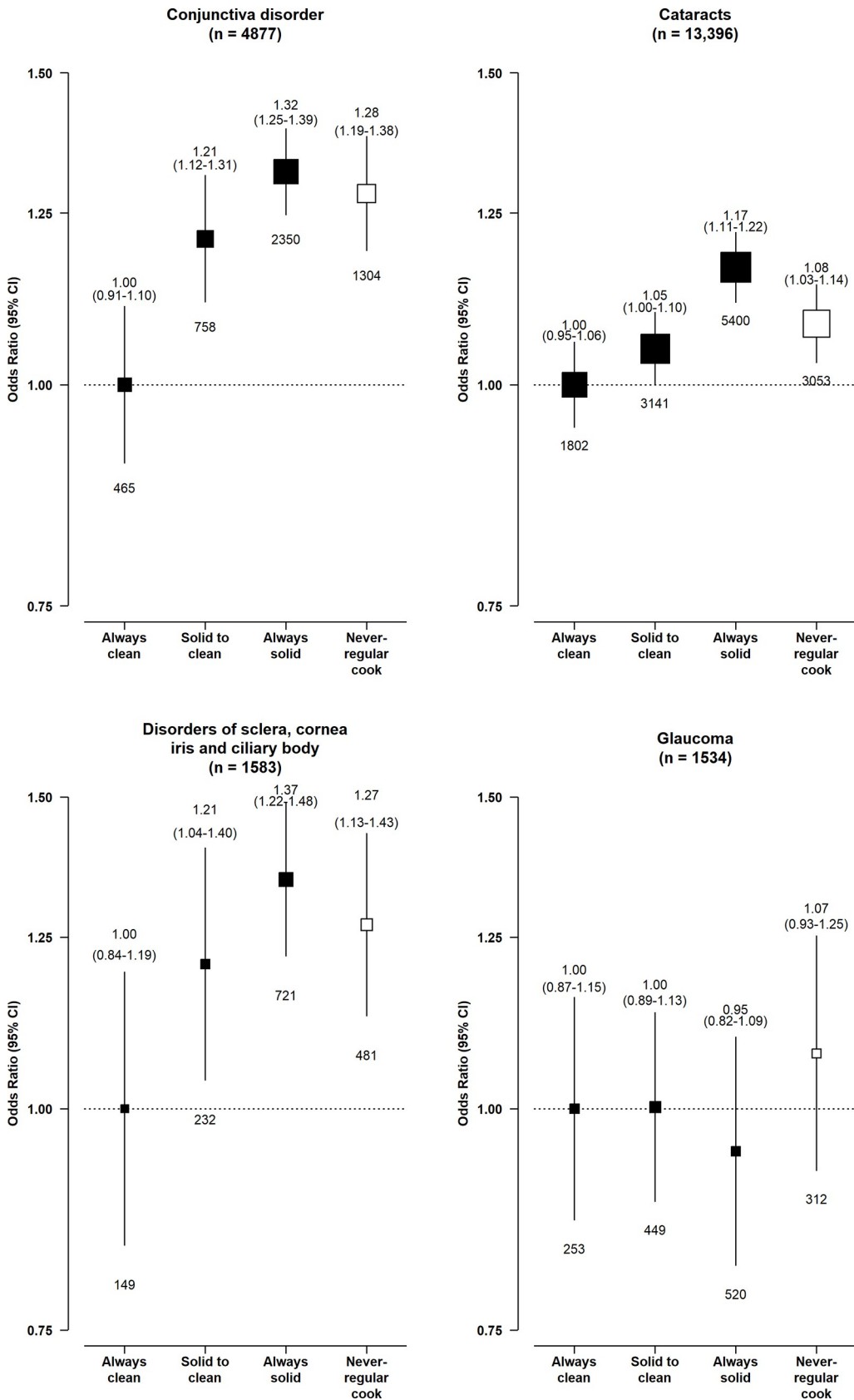

**Fig 1. Associations of long-term cooking fuel exposure with risk of major eye disease.** ORs were adjusted for age at baseline, birth cohort, sex, study area, education, occupation, smoking, environmental tobacco smoke, cookstove ventilation, heating fuel exposure, BMI, prevalent diabetes, self-reported general health, and length of recall period. The numbers in brackets are the total case number included in the 4 comparison groups for each disease endpoint. The boxes represent ORs, with the size inversely proportional to the variance of the logarithm of the category-specific log risk (which also determines the CIs represented by the vertical lines). The numbers above the vertical lines are point estimates and 95% CIs for ORs, and the numbers below the lines are numbers of events. Never-regular cook: individuals who reported cooking for monthly or less frequently throughout the recall period. BMI, body mass index; CI, confidence interval; OR, odds ratio.

highly heterogeneous, with reported ORs ranging from 0.4 [27] to >4.0 [28,29]. In a meta-analysis of 7 cross-sectional or case–control studies (involving a total of approximately 3,000 cataract cases) published during 1989 to 2005, the pooled OR was 2.5 (95% CI 1.74 to 3.50; $I^2$ = 62%) for "exposed" compared to "non-exposed" groups defined heterogeneously across studies [8]. Recent reports from several larger studies (with some involving up to approximately 4,000 cases [9]) found a smaller elevated risk of cataracts compared to earlier studies [9,30–32]

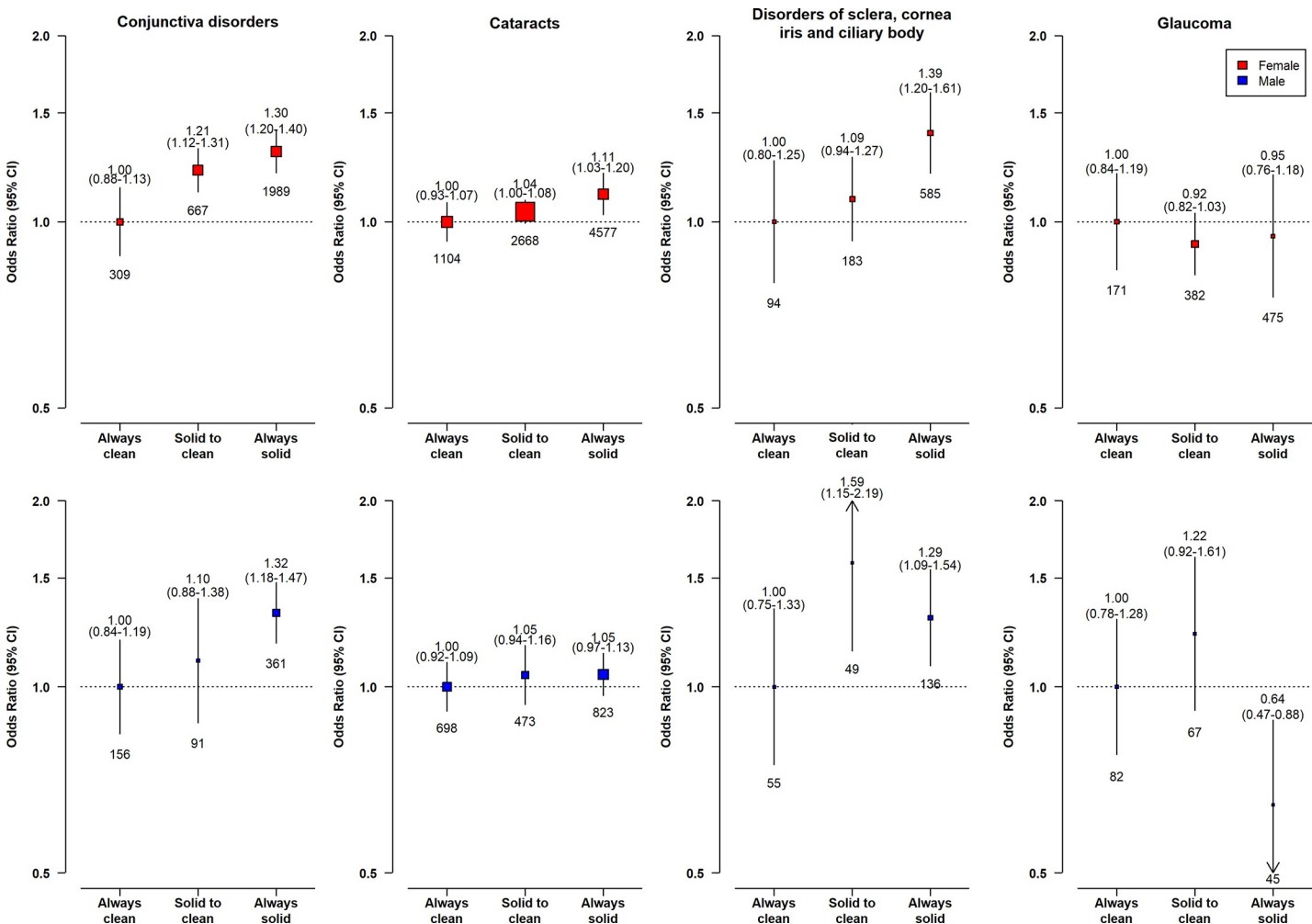

**Fig 2. Associations of long-term cooking fuel exposure with for major eye disease incidence in female (red) and male (blue).** ORs were adjusted for age at baseline, birth cohort, study area, education, occupation, smoking, environmental tobacco smoke, cookstove ventilation, heating fuel exposure, BMI, prevalent diabetes, self-reported general health, and length of recall period. The graphics are formatted as in Fig 1. BMI, body mass index; CI, confidence interval; OR, odds ratio.

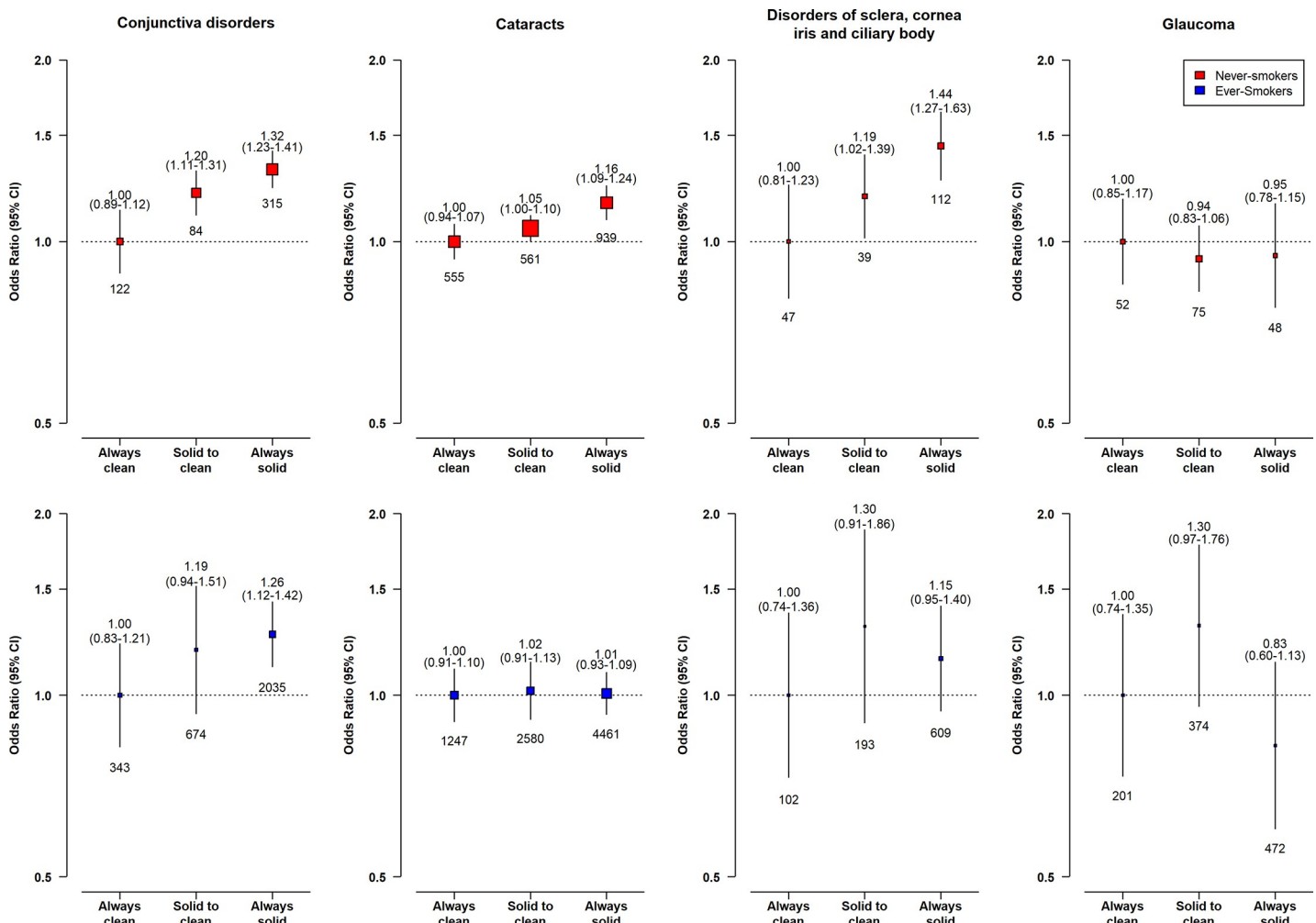

**Fig 3. Associations of long-term cooking fuel exposure with major eye disease incidence in never- (red) and ever- (blue) regular smokers.** ORs were adjusted for age at baseline, birth cohort, sex, study area, education, occupation, passive smoking, cookstove ventilation, heating fuel exposure, BMI, prevalent diabetes, self-reported general health, and length of recall period. The graphics are formatted as in Fig 1. BMI, body mass index; CI, confidence interval; OR, odds ratio.

(Table I in **S1 Tables**). In particular, a recent large cross-sectional study involving >4,000 cataract cases in India found an 18% (OR = 1.18, 1.02 to 1.36) higher risk in women and no association in men (1.04, 0.88 to 1.23) per 10 years longer household use of biomass for cooking [9]. In the present cohort study with larger number (>13,000) of cataract cases, we observed a 17% higher risk of cataracts comparing long-term solid fuel with clean fuel users, which is concordant with this Indian study. These suggested that the disease burden of cataracts attributed to solid fuel use for cooking may have been overestimated.

As in the Indian study [9], the present study also found that the elevated risk of cataracts associated with solid fuel use was mainly limited to women. It is likely that the sex difference in risk may be attributed to women's traditional role in cooking in LMIC settings, which entails substantially higher household air pollution exposure compared to men in the same household using solid fuels [33]. Unlike most previous studies that assessed only household fuel or stove types in women (because of presumptions on sex roles in cooking), we assessed the exposure by considering personal cooking frequency and included both men and women. At baseline, only 56% male regular cooks cooked daily, compared to >91% in female regular

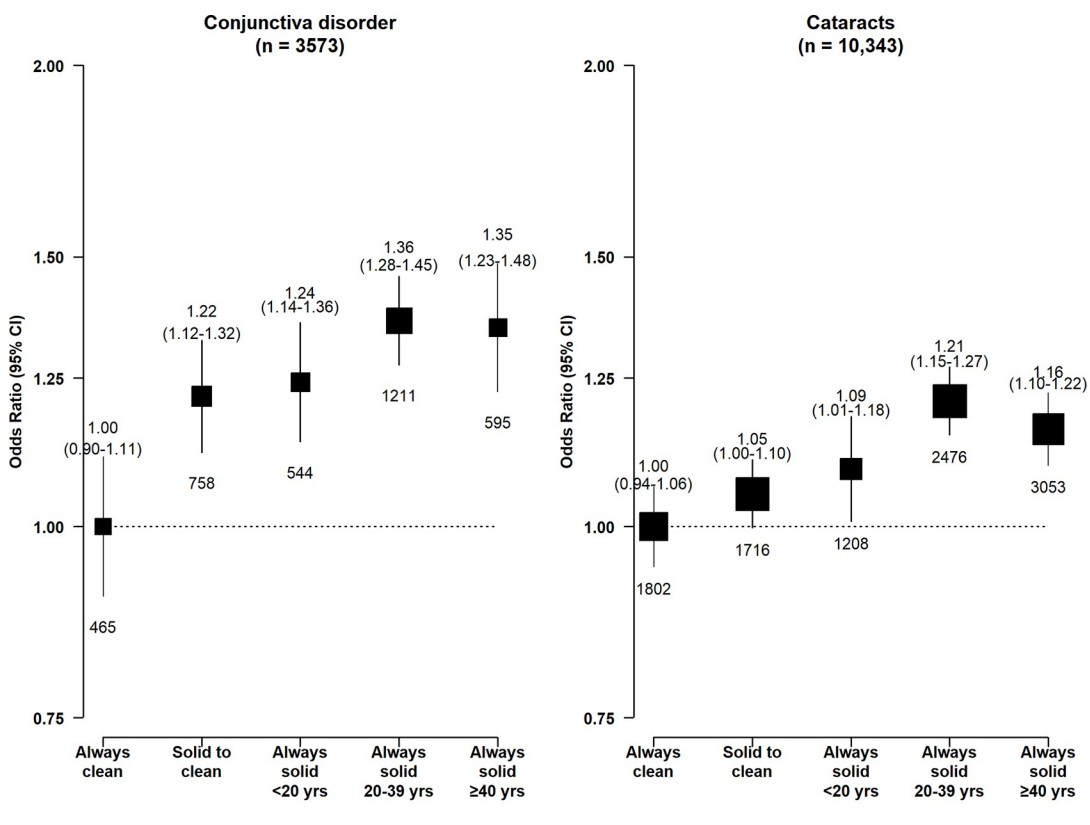

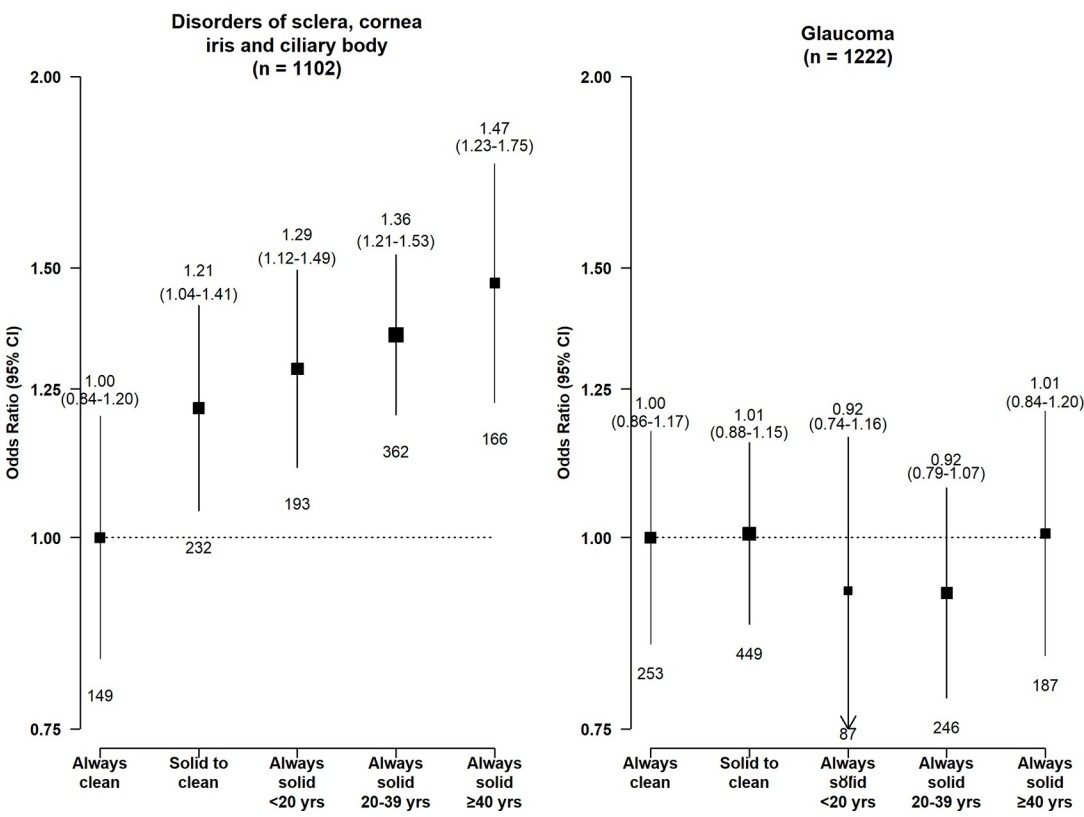

**Fig 4. Associations of duration of solid fuel use with risk of major eye disease.** ORs were adjusted for age at baseline, birth cohort, sex, study area, education, occupation, smoking, environmental tobacco smoke, cookstove ventilation, heating fuel exposure, BMI, prevalent diabetes, and self-reported general health. The numbers in brackets are the total case number included in the 5 comparison groups for each disease endpoint. The graphics are formatted as in Fig 1. BMI, body mass index; CI, confidence interval; OR, odds ratio.

cooks. Although more detailed cooking behaviour was not assessed at baseline, in a recent air pollution exposure measurement study involving 477 individuals in CKB, the mean daily cooking duration reported by male regular cooks was 0.45 h (95% CI 0.21 to 0.70, $P_{t\ test} <$ 0.001) shorter than female regular cooks [34]. Moreover, it is known that women are inherently at higher risk of cataracts [35], so it is possible that household air pollution is particularly harmful for women due to their underlying risk profile. However, the observed sex difference may also be due partly to play of chance because of the lower case numbers in the relatively small number of male regular cooks in CKB. Similarly, the apparent multiplicative interaction between solid fuel use and smoking in relation to cataracts may be explained by the potential elevated background risk in smokers, but it may also be reflecting the stark sex difference in smoking habits in the study population, with smokers being predominantly men (94%).

Nonspecific eye symptoms (e.g., redness, tears, dryness, pain, and irritation) are some of the most commonly assessed ocular outcomes in previous epidemiological studies on household air pollution, perhaps due to their ease of ascertainment compared to clinical eye diseases [4]. Generally, solid fuel use is associated with higher prevalence of self-reported eye symptoms, but their subjective and heterogeneous nature leave ambiguity about the relevance of solid fuel use to more severe forms or types of eye diseases, especially those requiring secondary care [4]. Although nonspecific, these symptoms are closely linked to DSCIC and conjunctiva disorders, most commonly conjunctivitis—one of the most prevalent eye diseases worldwide. In this large cohort study, we found evidence of elevated risk (32%, 95% CI 7% to 37%) of conjunctiva disorders in long-term solid fuels users, corroborating previous evidence on eye symptoms. Despite being usually self-limiting, the high occurrence and recurrent nature of conjunctivitis and the associated loss of productivity predispose to profound public health and economic burden (e.g., USD 800 million annually in the United States) [36]. Regretfully, little reliable estimates exist on the disease burden attributed to conjunctiva disorders in LMICs, where the impact is likely to be disproportionately larger than in high-income countries. Nonetheless, should our observation be verified in future epidemiological investigations, the global health impact of household air pollution from solid fuel use would be significantly higher.

No previous studies have examined the risks of DSCIC associated with solid fuel use. DSCIC is a group of relatively severe diseases of anterior and superficial structures of the eyes (other than the lens and conjunctiva) that are potentially susceptible to the harm of solid fuel smoke. The present study explored the association and provided novel epidemiological evidence supporting a link between solid fuel use and DSCIC. Of the 1,583 cases recorded in the present study, most were either keratitis (72.7%) or iridocyclitis (16.0%), inflammation of the cornea or iris and ciliary body of the eyes, respectively, both of which are important causes of vision impairment and blindness [37]. While each type of DSCIC has its own distinctive characteristics, they are known to be linked to conjunctiva disorders, and a strong association is observed between the 2 disease entities in the present study sample (adjusted OR = 7.11 [6.14 to 8.22]). Given the association of solid fuel use with conjunctiva disorders, it may act through common pro-inflammation mechanisms or via increasing the risk of conjunctiva disorders through keratitis or iridocyclitis. Another plausible pathway is that burning or handling of solid fuels, especially wood, may increase the chance of anterior eye injuries (a risk factor of

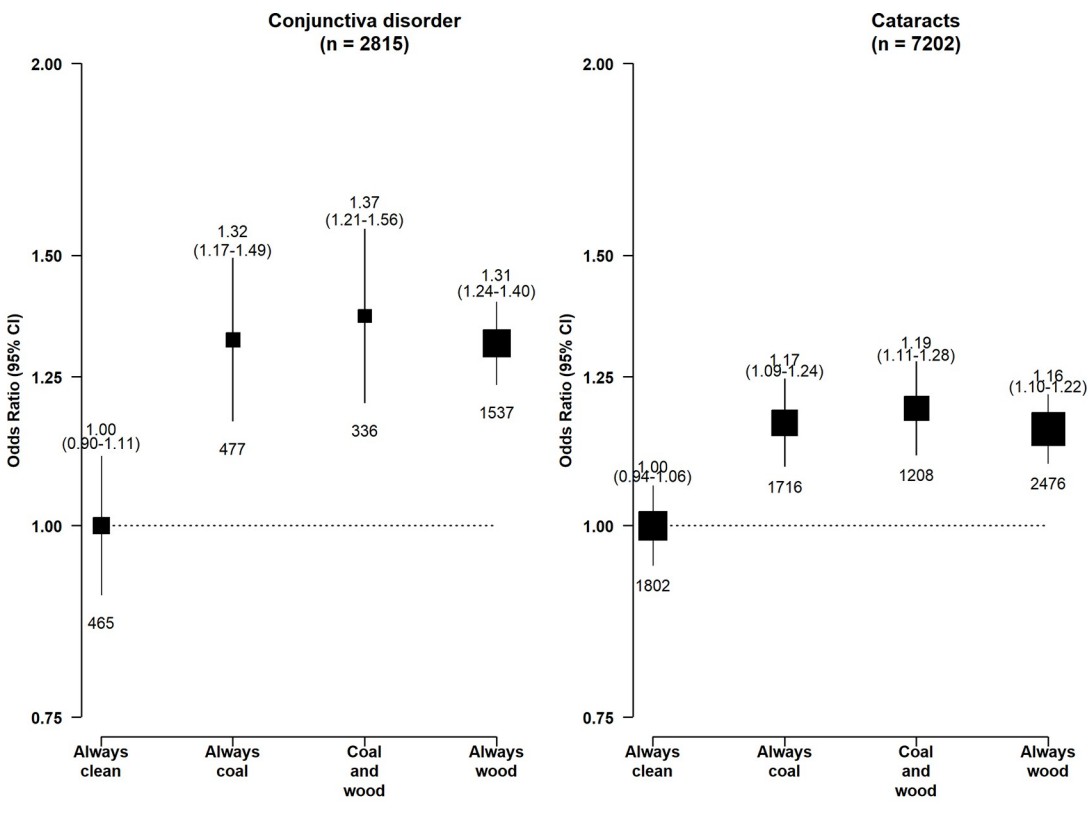

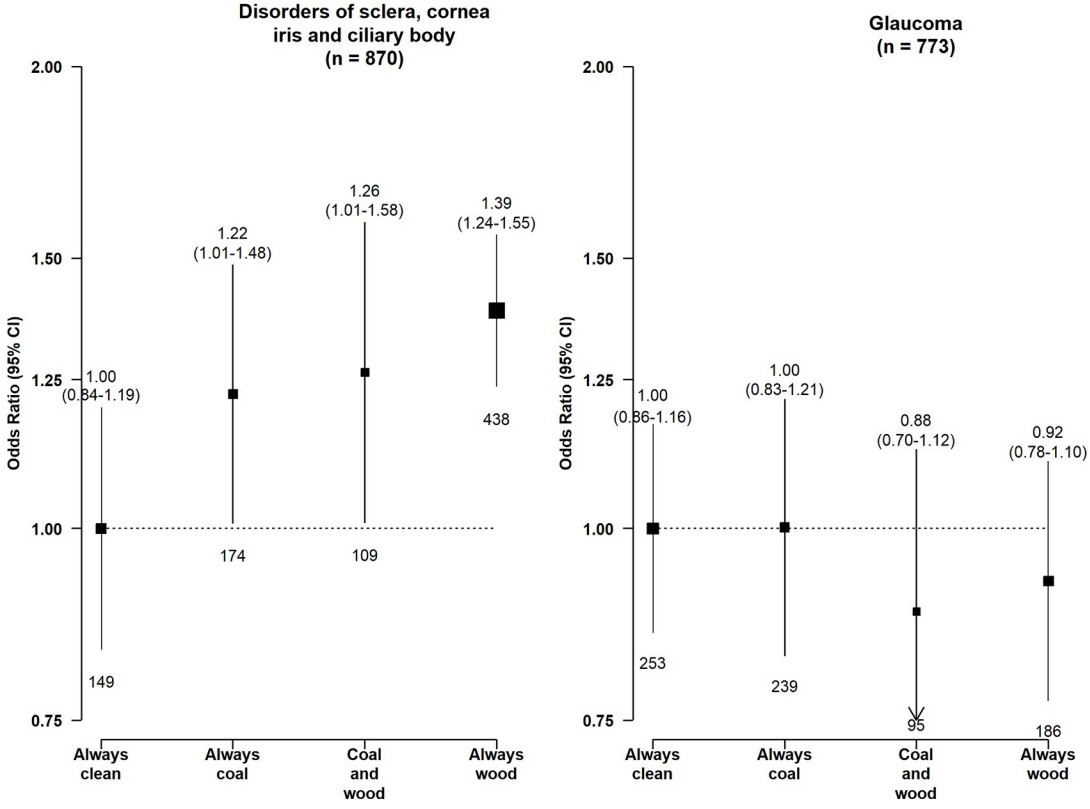

**Fig 5. Associations of use of specific solid fuel types with risk of major eye disease.** The adjustment employed for the ORs and the graphical format were the same as in Fig 1. The numbers in brackets are the total case number included in the 4 comparison groups for each disease endpoint. CI, confidence interval; OR, odds ratio.

DSCIC, particularly keratitis) from sparks or wood dust, which may explain why long-term wood users appeared to be at considerably higher risk (OR = 1.39 versus 1.22 in coal users) in our study. Despite the relatively large sample size, our study lacked the power to investigate the associations of solid fuel use with each of the specific DSCIC, which have heterogeneous pathophysiology and may not necessarily be subject to the same impact from household air pollution. In the absence of previous studies on household air pollution and DSCIC, our study has generated a new hypothesis that warrants further investigation on the association of solid fuel use with each of the specific DSCIC.

We found no relevant epidemiological study examining the association of solid fuel use and glaucoma, but a recent cross-sectional analysis on ambient air pollution in >111,000 UK Biobank participants reported a marginally significant 6% (95% CI 1% to 12%) higher risk of self-reported prior diagnosis of glaucoma per 1 $\mu g/m^3$ higher exposure to ambient $PM_{2.5}$, yet found no association with intraocular pressure (IOP) measured at the baseline assessment [38]. Interestingly, we found no evidence of an elevated risk of glaucoma in solid fuel users, despite the fact that solid fuel use is associated with 10- to 100-fold higher exposure to $PM_{2.5}$ than the above study [34,39]. Notably, the aetiology of glaucoma remains poorly understood, and most established risk factors are nonmodifiable (e.g., age, history of other eye diseases, and genetic factors) [40]. Unlike the other outcomes studied, glaucoma concerns the internal structure of the eyes, and the predominant subtype in Chinese are acute-closure glaucoma as opposed to the open-angle subtype in Western populations [41]. While it is plausible that air pollutants can reach the aqueous humour through the cardiorespiratory system and increase IOP by blocking the circulation, the previously reported null association between ambient $PM_{2.5}$ and IOP offered counter evidence [38]. It is possible that much of the systemic effects of household air pollution are "consumed" by the circulatory and hepatic systems, as we have previously demonstrated that solid fuel use is linked to major cardiovascular and hepatic diseases [15,42]. The null association observed for glaucoma (which is strongly linked to other eye diseases, particularly DSCIC, in our study) in the present study also suggests that the associations of solid fuel use with other outcomes are unlikely to be driven by the mutual correlation between different eye diseases.

The potential mechanisms of household air pollution exposure and eye diseases are not clearly understood and may vary by disease [43]. The primary pollutant in solid fuel smoke is $PM_{2.5}$, a mixture of thousands of noxious chemicals including polycyclic aromatic hydrocarbons and heavy metals [44]. $PM_{2.5}$ is known to induce oxidative stress and inflammation in the respiratory and cardiovascular systems [44,45] and increase the risks of both upper and lower respiratory infections, possibly through hampering the respiratory immune response [13]. Therefore, it is highly plausible that solid fuel smoke can also deposit on the eyes and alter the chemical equilibrium and immunity of the tear film, thus increasing the risk of infection and damaging ocular cells directly [2]. The free radicals in solid fuel smoke may accelerate the oxidation of the lens leading to cataracts [46]. Carbon monoxide, another prominent pollutant generated from incomplete combustion of solid fuels, may harm the eyes through hypoxia [47]. Future investigation into the chemical composition of tear or aqueous humour samples from solid fuel users may offer important insight into the potential pathogenesis pathways.

Previous intervention studies (mostly nonrandomised) on household air pollution have shown somewhat consistent evidence of reduced eye symptoms and conjunctivitis in those who adopted clean fuels or improved ventilation, but most of these studies suffered from

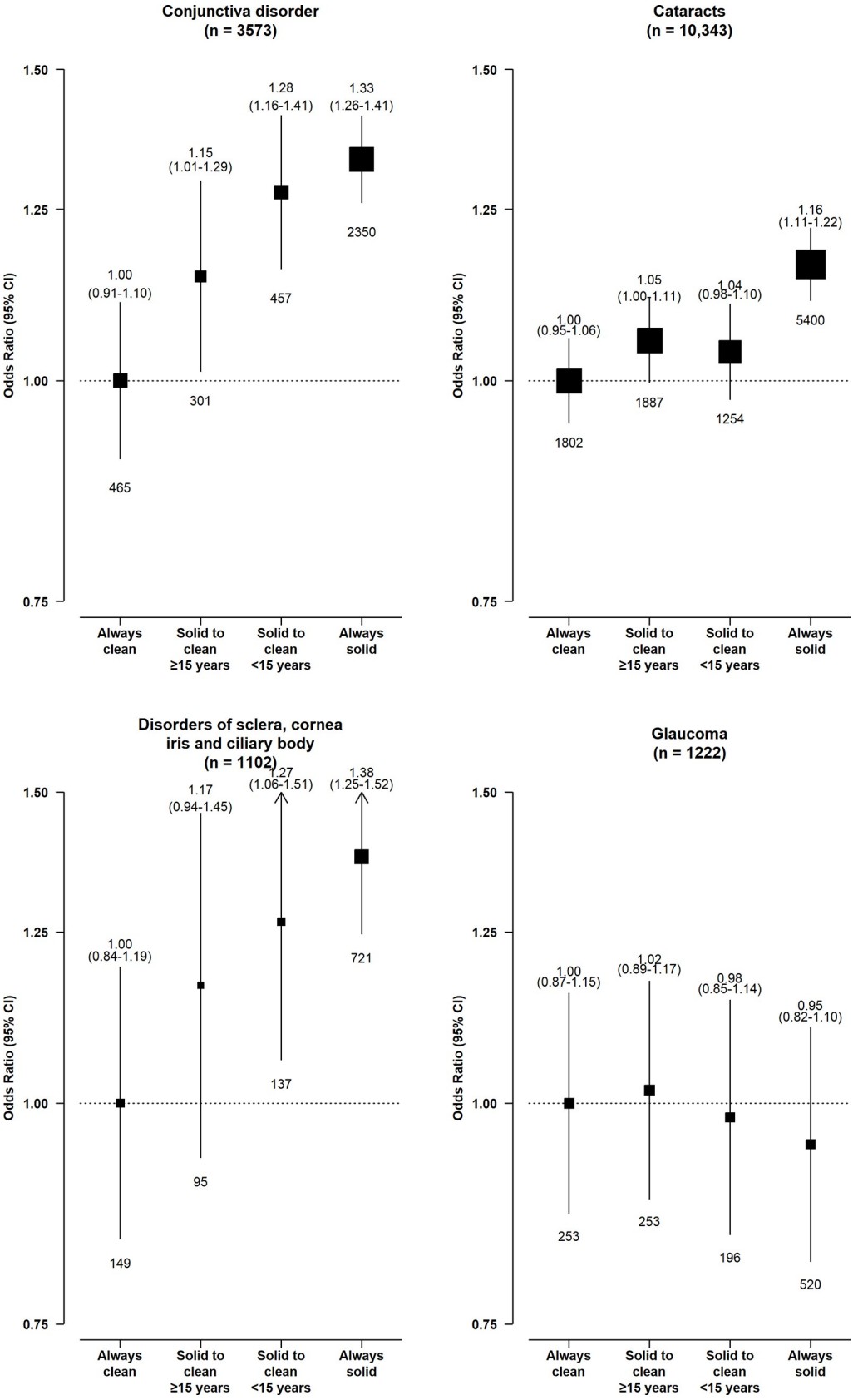

**Fig 6. Associations of clean fuel adoption with risks of major eye disease.** The adjustment employed for the ORs and the graphical format were the same as in Fig 1. The numbers in brackets are the total case number included in the 4 comparison groups for each disease endpoint. CI, confidence interval; OR, odds ratio.

major methodological limitations including noncompliance, cross-contamination, poor reporting of methods and results, residual confounding, and small sample size [3,48]. We found suggestive evidence that switching from solid to clean fuels is associated with lower risks of conjunctiva disorders, cataracts, and DSCIC compared to long-term solid fuel users, with indication of lower risks associated with earlier switching. However, we observed no evidence of benefit from better cookstove ventilation. One possible explanation for such a contrast is that clean fuel adoption reduces household air pollution exposure more substantially than ventilation, and the household air pollution levels in solid fuels–using households with ventilation often remain high (mean kitchen $PM_{2.5}$ concentration approximately 380 μg/m$^3$ versus approximately 120 μg/m$^3$ for ethanol stoves from a meta-analysis) [49]. Moreover, it is possible that some solid fuel–using households with worse household air pollution are more likely to install ventilation as a low-cost mitigation strategy, as observed in a small subset of CKB participants [34]. The heterogeneous nature and unknown effectiveness of cookstove ventilation in the study population may have introduced further noise to the analysis, masking any true association.

The strengths of this study are the large and diverse population, enhanced exposure assessment (incorporating fuel types and cooking behaviour), and systematic investigation of several understudied eye diseases. There are also several key limitations in our study. First, despite the enhancement in exposure assessment (combining personal cooking frequency and primary fuel type), it was not feasible to collect objectively measured household air pollution exposure data in the entire cohort, and we had no information on household fuel use among never-regular cooks. It is possible that historical or concurrent exposure to household air pollution from secondary or neighbourhood fuels have elevated the background risk of eye disease in primary clean fuel users, and this could have diluted the associations examined. The lack of objective exposure data also prevented us from directly assessing the shape of the dose–response relationships, although the findings on duration of exposure have offered some insight. Furthermore, it is recognised that non-cooking individuals who live in households using solid fuels for cooking may also be exposed to household air pollution [50], so the comparison of eye disease risk between never-regular cooks and regular cooks grouped by personal fuel use status must be interpreted with caution. Second, the lack of baseline eye examination prevented us from excluding individuals with preexisting conditions, so some events may simply be delayed diagnosis or treatment of such conditions. Serious eye conditions such as cataracts, aphakia, some forms of DSCIC, and glaucoma may stop people from cooking (thus reducing exposure) or prompt switching from solid to clean fuels. This reverse causation could dilute the associations by reducing the risk in the exposed group or inflate the risk in the "switcher" group. Although longer duration of exposure appeared to be associated with higher risk of cataracts, the risk in participants exposed for ≥40 years was similar to those exposed for 20 to 39 years. This may reflect a higher proportion of older individuals in the longer exposure group (mean age 60 years versus 51 years), who may already have had a cataract operation prior to baseline and were no longer at risk of cataracts. This may have underestimated the real association between household air pollution and cataracts and glaucoma, and to a lesser extent, other relatively acute conditions. We attempted to assess the extent of such biases from individuals with preexisting eye conditions in the sensitivity analyses excluding elderly (aged ≥65 years at baseline), who should have accounted for the majority of preexisting cataract cases [51], and the

first 3 years of follow-up, a reasonably long period that the subsequent events, particularly the acute outcomes (i.e., conjunctiva disorders and DSCIC), are less influenced by previous events at baseline. These analyses showed no material changes in the results, but the risk of bias remains an important issue of concern. Third, it was not possible for us to conduct regular standardised clinical eye examinations (as in some previous studies [9,27,52]) during follow-up. Since delays in diagnosis of eye disease, particularly cataracts, are common in LMICs, relying on routine health insurance records for outcome assessment may bias the associations towards the null. The use of the national health insurance data also constrained the study to more severe eye disease events requiring treatment in hospitals or health insurance reimbursement (as opposed to simple over-the-counter drug treatment) and omitted other potentially prevalent conditions such as dry eye disease [53,54], which has been linked to solid fuel use in previous studies [55]. It is also possible that patients with mild dry eye disease were misclassified as having conjunctivitis because dry eye disease could be secondary to conjunctivitis and they usually share some common symptoms (e.g., redness, itchiness, and stinging), especially given the general lack of objective or laboratory-confirmed diagnoses in China [54]. Furthermore, detailed information of cataract subtypes was not captured in the health insurance databases, so further analysis by subtypes was not possible. Fourth, despite the extensive adjustment for a range of potential confounders, residual confounding from SES or smoking (due to reporting bias) or unmeasured confounders (e.g., sunlight, occupational dust, heat from firepower, or ambient air pollution exposure) may still remain. For example, it is possible that individuals who used solid fuels, who were more likely to be agricultural workers in CKB, were exposed to more dust particles and sunlight, which are potential risk factors for the eye diseases examined [56,57], and the associations may be overestimated due to residual confounding. We adjusted for proxy exposures, including occupation, study areas, and physical activity levels in the regression models, but residual confounding is still likely. Heat exposure related to cooking is another potential confounder for eye disease (particularly cataracts [58,59]), although the relevant epidemiological evidence is scarce and little direct data exist to compare heat exposure in the eyes of solid fuels to clean fuels users. Further epidemiological studies measuring not only household air pollution but also heat exposure to the eyes would help to tease out their independent associations with eye disease. Overall, given the relatively modest ORs observed and the large sample size, caution is required in the interpretation of these results due to residual confounding.

In summary the present study provided new evidence linking long-term household air pollution exposure from solid fuel use with higher risks of major eye diseases (conjunctiva disorders, cataracts, and DSCIC) in a Chinese population. The associations appeared similar for wood and coal use and were largely independent of smoking and other risk factors. For cataracts, though statistically significant, especially among women, the risk estimates were more modest compared with those shown in earlier reports based on relatively small case–control or cross-sectional studies, corroborating with the more recent, large-scale investigations. In addition, the results suggested the potential benefits of switching from solid to clean fuels, underscoring the value of promoting access to clean and affordable household energy worldwide. Future studies employing regular and standardised eye examination in a large prospective cohort, along with enhanced household air pollution exposure assessment and comprehensive coverage of confounders, are warranted to further clarify the impact of solid fuel use on eye health, especially to directly assess temporality and also examine milder eye diseases.

## Supporting information

**S1 STROBE Checklist. STROBE checklist.**
(DOCX)

**S1 Text. Supplementary methods.**
(DOCX)

**S1 Tables.** Table A. Major categories of eye disease examined. Table B. Associations between the major eye diseases examined. Table C. Odds ratios and group-specific 95% confidence intervals for major eye diseases according to long-term cooking fuel use—results of sensitivity analysis (1). Table D. Odds ratios and 95% confidence intervals for major eye diseases according to long-term cooking fuel use—results of sensitivity analysis (2). Table E. Odds ratios and 95% confidence intervals for major eye diseases in long-term solid fuel users versus clean fuel users from leave-one-out sensitivity analysis. Table F. Comparison of odds ratios (ORs) of primary analysis and hazard ratios (HRs) estimates from Cox regression analysis. Table G. Comparison of odds ratios (ORs) of primary analysis on duration of solid fuel use and hazard ratios (HRs) estimates from Cox regression analysis. Table H. Comparison of odds ratios (ORs) of primary analysis on types of solid fuel use and hazard ratios (HRs) estimates from Cox regression analysis. Table I. Characteristics of cross-sectional and case–control studies evaluating household air pollution and the risk of cataract.
(DOCX)

**S1 Figs.** Fig A. Graphical illustration of potential bias from the disproportionately delayed treatment or diagnosis in solid fuel users. Fig B. Associations of cookstove ventilation availability with major eye disease incidence in long-term solid fuel users
(DOCX)

# Acknowledgments

The chief acknowledgment is to the participants, the project staff, and the China National Center for Disease Control and Prevention (CDC) and its regional offices for assisting with the fieldwork. We thank Judith Mackay in Hong Kong; Yu Wang, Gonghuan Yang, Zhengfu Qiang, Lin Feng, Maigeng Zhou, Wenhua Zhao, and Yan Zhang in China CDC; Lingzhi Kong, Xiucheng Yu, and Kun Li in the Chinese Ministry of Health; and Garry Lancaster, Sarah Clark, Martin Radley, Mike Hill, Hongchao Pan, and Jill Boreham in the CTSU, Oxford, for assisting with the design, planning, organisation, and conduct of the study.

The members of the China Kadoorie Biobank collaborative group are as follows:

**International Steering Committee:** Junshi Chen, Zhengming Chen (PI), Robert Clarke, Rory Collins, Yu Guo, Liming Li (PI), Chen Wang, Jun Lv, Richard Peto, Robin Walters.

**International Co-ordinating Centre, Oxford:** Daniel Avery, Ruth Boxall, Derrick Bennett, Ka Hung Chan, Yumei Chang, Yiping Chen, Zhengming Chen, Robert Clarke, Huaidong Du, Zammy Fairhurst-Hunter, Wei Gan, Simon Gilbert, Alex Hacker, Parisa Hariri, Mike Hill, Michael Holmes, Pek Kei Im, Andri Iona, Maria Kakkoura, Christiana Kartsonaki, Rene Kerosi, Kuang Lin, John McDonnell, Iona Millwood, Qunhua Nie, Alfred Pozarickij, Paul Ryder, Sam Sansome, Dan Schmidt, Paul Sherliker, Rajani Sohoni, Becky Stevens, Iain Turnbull, Robin Walters, Lin Wang, Neil Wright, Ling Yang, Xiaoming Yang, Pang Yao.

**National Co-ordinating Centre, Beijing:** Zheng Bian, Yu Guo, Xiao Han, Can Hou, Chun Li, Chao Liu, Jun Lv, Pei Pei, Canqing Yu.

10 Regional Co-ordinating Centres:

**Guangxi** Provincial CDC: Naying Chen, Duo Liu, Zhenzhu Tang. Liuzhou CDC: Ningyu Chen, Qilian Jiang, Jian Lan, Mingqiang Li, Yun Liu, Fanwen Meng, Jinhuai Meng, Rong Pan, Yulu Qin, Ping Wang, Sisi Wang, Liuping Wei, Liyuan Zhou. **Gansu** Provincial CDC: Caixia Dong, Pengfei Ge, Xiaolan Ren. Maiji CDC: Zhongxiao Li, Enke Mao, Tao Wang, Hui Zhang, Xi Zhang. **Hainan** Provincial CDC: Jinyan Chen, Ximin Hu, Xiaohuan Wang. Meilan CDC:

Zhendong Guo, Huimei Li, Yilei Li, Min Weng, Shukuan Wu. **Heilongjiang** Provincial CDC: Shichun Yan, Mingyuan Zou, Xue Zhou. Nangang CDC: Ziyan Guo, Quan Kang, Yanjie Li, Bo Yu, Qinai Xu. **Henan** Provincial CDC: Liang Chang, Lei Fan, Shixian Feng, Ding Zhang, Gang Zhou. Huixian CDC: Yulian Gao, Tianyou He, Pan He, Chen Hu, Huarong Sun, Xukui Zhang. **Hunan** Provincial CDC: Biyun Chen, Zhongxi Fu, Yuelong Huang, Huilin Liu, Qiaohua Xu, Li Yin. Liuyang CDC: Huajun Long, Xin Xu, Hao Zhang, Libo Zhang. **Jiangsu** Provincial CDC: Jian Su, Ran Tao, Ming Wu, Jie Yang, Jinyi Zhou, Yonglin Zhou. Suzhou CDC: Yihe Hu, Yujie Hua, Jianrong Jin Fang Liu, Jingchao Liu, Yan Lu, Liangcai Ma, Aiyu Tang, Jun Zhang. **Qingdao** Qingdao CDC: Liang Cheng, Ranran Du, Ruqin Gao, Feifei Li, Shanpeng Li, Yongmei Liu, Feng Ning, Zengchang Pang, Xiaohui Sun, Xiaocao Tian, Shaojie Wang, Yaoming Zhai, Hua Zhang, Licang CDC: Wei Hou, Silu Lv, Junzheng Wang. **Sichuan** Provincial CDC: Xiaoyu Chang, Xiaofang Chen, Xianping Wu, Ningmei Zhang. Pengzhou CDC: Xiaofang Chen, Jianguo Li, Jiaqiu Liu, Guojin Luo, Qiang Sun, Xunfu Zhong. **Zhejiang** Provincial CDC: Weiwei Gong, Ruying Hu, Hao Wang, Meng Wan, Min Yu. Tongxiang CDC: Lingli Chen, Qijun Gu, Dongxia Pan, Chunmei Wang, Kaixu Xie, Xiaoyi Zhang.

## Author Contributions

**Conceptualization:** Ka Hung Chan, Derrick A. Bennett, Huaidong Du, Kin Bong Hubert Lam, Zhengming Chen.

**Data curation:** Ka Hung Chan, Yiping Chen, Ling Yang.

**Formal analysis:** Ka Hung Chan.

**Funding acquisition:** Jun Lv, Liming Li, Kin Bong Hubert Lam, Zhengming Chen.

**Investigation:** Ka Hung Chan, Mingshu Yan, Canqing Yu.

**Methodology:** Ka Hung Chan, Jun Lv, Huaidong Du, Kin Bong Hubert Lam, Zhengming Chen.

**Project administration:** Yu Guo, Canqing Yu, Pei Pei, Yan Lu.

**Resources:** Yu Guo, Yiping Chen, Ling Yang, Jun Lv, Canqing Yu, Pei Pei, Liming Li.

**Software:** Canqing Yu.

**Supervision:** Derrick A. Bennett, Liming Li, Huaidong Du, Kin Bong Hubert Lam, Zhengming Chen.

**Visualization:** Ka Hung Chan, Mingshu Yan.

**Writing – original draft:** Ka Hung Chan.

**Writing – review & editing:** Ka Hung Chan, Mingshu Yan, Derrick A. Bennett, Yu Guo, Yiping Chen, Ling Yang, Jun Lv, Canqing Yu, Pei Pei, Yan Lu, Liming Li, Huaidong Du, Kin Bong Hubert Lam, Zhengming Chen.

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
