## [Editor Report · Decision Letter 0]

25 Sep 2020

Dear Dr Lam, 

Thank you for submitting your manuscript entitled "Long-term solid fuel use and risks of major eye diseases: a 10-year prospective cohort study of 0.5 million adults in China" for consideration by PLOS Medicine.

Your manuscript has now been evaluated by the PLOS Medicine editorial staff and I am writing to let you know that we would like to send your submission out for external assessment.

Sincerely,

Richard Turner, PhD

Senior editor, PLOS Medicine

rturner@plos.org

---

## [Decision Letter · Decision Letter 1]

2 Nov 2020

Dear Dr. Lam,

Thank you very much for submitting your manuscript "Long-term solid fuel use and risks of major eye diseases: a 10-year prospective cohort study of 0.5 million adults in China" (PMEDICINE-D-20-04664R1) for consideration at PLOS Medicine. 

Your paper was evaluated by the editors and sent to independent reviewers, including a statistical reviewer. The reviews are appended at the bottom of this email and any accompanying reviewer attachments can be seen via the link below:

[LINK]

In light of these reviews, we will not be able to accept the manuscript for publication in the journal in its current form, but we would like to invite you to submit a revised version that addresses the reviewers' and editors' comments fully. You will appreciate that we cannot make a decision about publication until we have seen the revised manuscript and your response, and we expect to seek re-review by one or more of the reviewers. 

We hope to receive your revised manuscript by Nov 23 2020 11:59PM. Please email us (plosmedicine@plos.org) if you have any questions or concerns.

Please let me know if you have any questions. Otherwise, we look forward to receiving your revised manuscript in due course. 

Sincerely,

Richard Turner, PhD

rturner@plos.org

We think that you are reporting a retrospective analysis of a prospectively-gathered dataset. Therefore, please remove the word "prospective" from the title, and adapt mentions of this word throughout the text to refer to the parent study rather than the current analysis. 

We suggest moving "in China" before the colon in your title, with the study descriptor then becoming "a population-based cohort study", as noted elsewhere in your paper. 

In your abstract and throughout the text, please quote p values alongside 95% CI, where available. 

Please make that "A questionnaire ..." in your abstract.

Please add a new final sentence to the "Methods and findings" subsection of your abstract, quoting 2-3 of the study's main limitations. 

After the abstract, we will need to ask you to add a new and accessible "Author summary" section in non-identical prose. You may find it helpful to take a look at one or two recent research papers in PLOS Medicine to get a sense of the preferred style. 

Please remove the information on funding from the title page, the "Role of the funding source" statement from the end of the methods section, and the information on funding and competing interests from the end of the main text. In the event of publication, this information will appear in the article metadata, via entries in the submission form. 

Early in the methods section of your main text, please state whether the present analysis had a protocol or prespecified analysis plan, and if so attach the relevant document(s) as a supplementary file(s) in the text. Please highlight analyses that were not prespecified, including any carried out in response to referees' comments. 

"Written informed consent" is mentioned twice early in the methods section, and once will suffice. 

Throughout the text, please adapt reference call-outs to the following style: "... and their families [5,6]." (i.e., preceding punctuation and with no spaces in the square brackets). 

In your reference list, please ensure that journal names are abbreviated consistently (e.g., "Lancet" and derivatives). 

Please update reference 25, or supply the latest version with your revision. 

Please remove the competing interest information from reference 32. 

Please add a completed checklist for the most appropriate reporting guideline, which we suspect we will be STROBE, as a supplementary document, referred to in the methods section ("See S1_STROBE_Checklist" or similar). In the checklist, please refer to individual items by section (e.g., "Methods") and paragraph number rather than by line or page numbers, as the latter generally change in the event of publication. 

Comments from the reviewers:

*** Reviewer #1: 

I mostly confine my remarks to statistical aspects of this paper. One note is that it could use some editing for English usage, particularly the abstract.

On to statistics; the general method is fine, but I have some questions and issues to resolve before I can recommend publication.

p. 2 - "Not alter the risks" .... not at all? Not signficcantly? Not in any clinically important way? Or what?

p. 6 - Categorizing continuous variables is a bad idea. If the data on age was collected this way then there is not much you can do (but it should be acknowledged) but if age was collected as "years" then leave it that way and look at a spline to examine nonlinearity

p. 7- Please give some details of how "direct standardization" was done. 

Tables - I commend the authors for leaving out p values. They really wouldn't add much.

Peter Flom

*** Reviewer #2: 

This study investigated the long-term adverse effects of solid fuel use for cooking on major ocular diseases. This is an excellent study analyzed with a large, long-term prospective cohort. In particular, I think this especially provided important evidence for cataracts and conjunctival disorders. However, there are some issues in this study.

1. It is possible that UV exposure or outdoor air pollution has influenced the results. What do you think of the impact on these factors? Do you think the current analytic methods already has made relevant adjustments?

2. Why is the OR of 'never cook' as high as solid to clean? Shouldn't it be similar to control theoretically? Please add to discussion.

3. Dry eye disease is the most common ocular disease most likely to be affected by household air pollution. Is dry eye disease included in a conjunctival disorder or DSCIC in the current analysis? 

4. Retinal disease (macular degeneration or retinal vein occlusion) is one of the prevalent ocular disease. Why don't you include this?

5. People who cook on solid fuels would be exposed to relatively strong firepower compared to clean fuels. Heavy heat exposure has the potential to exacerbate several eye diseases, including cataracts. What do you think it is possible that strong heat exposure from solid fuels, not HAP, could have resulted in this?

6. Disorders of sclera, cornea, iris and ciliary body (DSCIC) is a wide range of diseases category and it has diverse heterogenous pathophysiology respectively, so I think these results are not helpful clinically or academically. It would be better to exclude the DSCIC part from the results.

*** Reviewer #3: 

This manuscript describing an analysis of the association between primary fuel type and risk of eye disorders within the CKB is generally well-written and contributes important information given the general lack of information on this topic. However, I have a few concerns described below.

1. Abstract - clarify the reference for solid fuels in methods and findings.

2. Page 7: Justify these exclusions further. Why not include these groups?

3. Page 7: Provide reference for statement about bias in conventional survival analysis. Explain this statement.

4. Page 7: further justification is needed for confounders (e.g., a DAG or additional information).

5. Page 7: what method was used to assess interaction? An interaction term or stratification? Clarify. Be clear in the text that only multiplicative interaction was assessed; incorporate this fact into the results interpretation (any information then about additive interaction?)

6. For interaction results it is not sufficient to just give p-value>0.05. Provide the p-value for each analysis. There actually seems to be a suggestion of interaction for cataracts by smoking status. 

7. Was there any loss to follow up in the cohort? Need to report.

8. A primary weakness of this study is the exposure assessment. Self reported primary fuel use is a crude assessment of exposure. On Page 16 this limitation needs further discussion - including the potential impact on the observed results.

9. A second important limitation is residual confounding. This potential limitation needs to be discussed further, including potential impact on the results. How likely is residual confounding? By sunlight? By agricultural exposures? By smoking (due to mismeasured smoking status)?

10. Provide additional details about ethics approvals. 

11. Table 1: provide both N and %

12. The label "Never-regular cook" is not clear; be sure to clarify in Figures and tables.

***

[LINK]

---

## [Decision Letter · Decision Letter 2]

31 Dec 2020

Dear Dr. Lam,

Thank you very much for submitting your revised manuscript "Long-term solid fuel use and risks of major eye diseases in China: a population-based cohort study of 0.5 million adults" (PMEDICINE-D-20-04664R2) for consideration at PLOS Medicine. 

Your paper was evaluated by the editors and sent to our independent reviewers, including a statistical reviewer. The reviews are appended at the bottom of this email and any accompanying reviewer attachments can be seen via the link below:

[LINK]

In light of these reviews, we will again be unable to accept the manuscript for publication in the journal in its current form, but we would like to invite you to submit a further revised version that fully addresses the reviewers' and editors' comments. You will note that we cannot make a decision about publication until we have seen the revised manuscript and your response, and we plan to seek re-review by one or more of the reviewers. 

We hope to receive your revised manuscript by Jan 28 2021 11:59PM. Please email us (plosmedicine@plos.org) if you have any questions or concerns.

Please let me know if you have any questions. Otherwise, we look forward to receiving your revised manuscript in due course. 

Sincerely,

Richard Turner, PhD

rturner@plos.org

Please revisit the wording at line 118 and any other instances, noting the further comments from one referee - we continue to view the study as a retrospective analysis of prospectively gathered data and ask you to adapt the language used to reflect the study design. 

Where appropriate, please substitute "p<0.001" for "p<0.0001" throughout the ms. 

Please substitute "sex" for "gender" throughout, again where appropriate. 

Comments from the reviewers:

*** Reviewer #1: 

The authors have addressed my concerns and I now recommend publication.

Peter Flom

*** Reviewer #2: 

I cannot find the evidence that there is no difference of heat exposure between solid and clean fuel in your reference. Solid fuel can expose a lot of heat to the eyes during the first ignition process or fire boosting. In addition, dry eye disease is the most prevalent disease in the world, and the known prevalence in the world is about 15-20%. UV or sunlight exposure should also be regarded as an independent risk factor in the ocular disease. Overall, I did not get an adequate answer to the question from you. I think there is a serious error in interpreting the results despite of large sample.

*** Reviewer #3: 

Thank you to the authors for a responsive revision and comments. I have a few remaining comments/suggestions. 

1. I see that the Editors requested p-values along with 95% CIs; I don't agree with this suggestion, but note that the authors did this to comply with the editors request. Please note the various recommendations throughout the statistical and epidemiologic literature to decrease emphasis on p-values. Perhaps this can be modified in the final version. Overall there is too much emphasis on statistical significance in the current version. In this regard, the sentence in the abstract about ventilation could be modified to state clearly that the ORs are similar regardless of ventilation, rather than the focus on the statistical significance (the statistical significance supports this result, it's not the primary result). 

2. I suggest not using the abbreviation HAP. Although common in this field, it also means hazardous air pollutants. 

3. I still have a concern about approach to confounding. There is no statistical test for confounding, despite what you sometimes see in the literature. The statistical tests are evaluating other aspects, not confounding. Once confounders have been identified (and DAGs or other methods used to make sure you are not adjusting for mediators or colliders) there is value perhaps to reducing the number of variables in the model (e.g., if removing them does not change the ORs in a meaningful way). There was also no reference cited for the approach. Please rely here on the epidemiologic approach to confounding, and not a statistical test. 

*** Reviewer #4: 

Although the authors have been somewhat responsive to criticisms, I have a few remaining concerns. 

1. The main one is that the study purports to be superior to previous studies because it is "prospective". In fact, a true prospective study would have carried out at least baseline eye examinations of the whole cohort and excluded those already with cataract, etc., from the multivariate models. It would then be possible to ascertain which of the diagnosed conditions prospectively occurred during the period of follow-up (preferably with periodic eye examinations of the whole cohort). The fact that baseline eye examinations did not occur means that the study more closely resembles a retrospective cohort study, in which a cohort is identified in the past and tracked into the future, using registries to identify diagnoses that have occurred since the start of the follow-up period. In such studies, there is always the possibility that the diseases of interest were present at the start of the follow-up period. The authors correctly identify the lack of baseline examinations as being a deficiency, but persist in calling it a prospective study (or at least implying that it is). Examples include:

* Abstract, line 31. "Little reliable prospective evidence exists…."

* Author summary, lines 68-69. "…but most previous studies used retrospectively collected data"

* Lines 86-87. "…this is the first investigation on the prospective relationship between long-term solid fuel use and risks of multiple common eye diseases."

* Lines 314-315 (in reference to previous studies): "Notably, all these studies were relatively small, cannot assess the temporality of association…"

* Line 323 (again, in reference to previous studies): "…they suffered from similar limitations, particularly the inability to assess temporality."

* Line 443. "The strengths of this study are the prospective follow-up…"

The authors reasonably assume, for SES reasons, that solid fuel users are likely to have a longer time to diagnosis than clean fuel users. This makes it even more essential that identification of baseline abnormalities and subsequent exclusions took place before claims of the superiority of their analysis can be made (examples above). They use the "sensitivity analyses" to argue that the lack of baseline examinations is not a problem, but this is a very indirect, weak and likely insensitive method of inference. The deficiency of uniform baseline (and subsequent) eye examinations is likely to impact logistic regression analysis as much as survival analysis (in contrast to their response to reviewer 3 and arguments on lines 201-208), and the bias is likely to be towards the null (assuming a HAP effect). The reason for this is that many solid fuel users may never be diagnosed within the follow up period of the study, or are certainly less likely to be diagnosed than higher SES clean fuel users.

2. My second concern is that the results and differences between males and females in terms of eye diseases is always of major interest and importance when comparing results between different studies, particularly with the HAP issue. I do not think this information should be buried in the supplementary material (Supplementary Figure 3). Also, all the supplementary figures should have the same information (i.e., confidence intervals) as the figures in the main paper.

3. I would make a similar argument for the interaction with smoking (Supplementary Figure 2), which deserves to be elevated from the supplementary material.

4. Why is BMI a covariate? What is the evidence that it may be a confounder—for all the eye diseases studied?

***

[LINK]

---

## [Decision Letter · Decision Letter 3]

16 Jun 2021

Dear Dr. Lam,

Thank you very much for re-submitting your manuscript "Long-term solid fuel use and risks of major eye diseases in China: a population-based cohort study of 0.5 million adults" (PMEDICINE-D-20-04664R3) for consideration at PLOS Medicine. We do apologize for the delay in sending you a response. 

I have discussed the paper with editorial colleagues and it was also seen again by three reviewers. I am pleased to tell you that, provided the remaining editorial and production issues are fully dealt with, we expect to be able to accept the paper for publication in the journal.

[LINK]

Please let me know if you have any questions, and we look forward to receiving the revised manuscript.   

Sincerely,

Richard Turner, PhD

rturner@plos.org

Requests from Editors:

We suggest addressing referee 2's second point by mentioning this issue as a possible limitation in the Discussion section (main text). 

Please quote the actual number of participants in the title rather than "0.5 million", and again at line 74.

We suggest mentioning the China Kadoorie Biobank in the abstract. 

Please quote the actual length of follow-up, as at line 259, in the abstract.

Please add a sentence, say, early in the "Methods and findings" subsection of your abstract to summarize the factors adjusted for.

At line 264, for example, please substitute "sex" for "gender" as appropriate.

Please use the journal name abbreviation "PLoS ONE" in the reference list.

Please break the STROBE checklist out into a separate attached file, labelled "S1_STROBE_Checklist" or similar, and refer to it by this label in the Methods section. 

Comments from Reviewers:

*** Reviewer #2: 

1) In the clinical aspects, conjunctival disorder, which you showed the major disorder related with solid fuel, are also relatively mild eye disease, similarly with dry eye disease. In addition, dry eye disease is more prevalent in East Asia (China, Korea, Japan, etc) In previous study using meta-analysis in China, pooled prevalence of dry eye disease in China was 17.0%. (Liu et al, J Ophthalmol. 2014;2014:748654.) In my opinion, if patients go to hospital due to conjunctival disorder frequently, patients also go to hospital due to dry eye disease similarly. If not, it is also possible that a patient with dry eye disease was incorrectly classified as a conjunctival disorder. In dry eye patients, conjunctival injections are very frequent. Therefore, I think this cohort data may not be reflect the real world situation adequately. Authors should emphasize strongly the possibility of underestimation or false classification of dry eye disease as a limitation.

2) In my previous review, I recommended to exclude DSCIC part from the results. Disorders of sclera, cornea, iris and ciliary body (DSCIC) is a wide range of diseases category and it has diverse heterogenous pathophysiology respectively, so I think these results cannot be helpful clinically or academically, even if I review it again now. Please exclude the results of DSCIC.

*** Reviewer #3: 

The authors have addressed my concerns. I continue to question BMI as a confounder given its potential role as a mediator; this potential role should at least be acknowledged. 

*** Reviewer #4: 

The authors have been appropriately responsive to comments and I have nothing further to add. The manuscript is ready for publication.

***

[LINK]

---

## [Editor Report · Decision Letter 4]

27 Jun 2021

Dear Dr. Lam,

Thank you very much for re-submitting your manuscript "Long-term solid fuel use and risks of major eye diseases in China: a population-based cohort study of 486,532 adults" (PMEDICINE-D-20-04664R4) for consideration at PLOS Medicine.

We will need to ask you to address some remaining issues, listed at the end of this email, before we are able to proceed further. 

In revising the manuscript for further consideration here, please ensure you address the specific points made by the editors. In your rebuttal letter you should indicate your response to the reviewers' and editors' comments and the changes you have made in the manuscript. Please submit a clean version of the paper as the main article file. A version with changes marked must also be uploaded as a marked up manuscript file.

Please let me know if you have any questions, and we look forward to receiving the revised manuscript shortly.   

Sincerely,

Richard Turner, PhD

rturner@plos.org

Requests from Editors:

Please adapt the data statement (submission form) to explain briefly the nature of restrictions on access to data - e.g., access can only be granted to investigators for stated research purposes. 

In the abstract, please make that "cataracts" (line 38) and "cases of glaucoma" (line 39), or similar, and adapt the phrasing similarly throughout the ms.

At line 94 (author summary) please revisit the statement "... appeared to be considerably weaker.". It appears that a few words need to be added to state the conditions under which the association is weaker. 

Please remove the information on study funding from the "Acknowledgements" at the end of the main text Information on funding should appear only in the article metadata, via entries in the submission form.

Please remove information on data access from the end of the main text. Again, this information should appear only in the article metadata.

Please remove the names of the CKB group from the end of the main text, and list these in a supplementary file. 

Please substitute "sex" for "gender" where appropriate throughout the ms, e.g., at line 42 in the abstract and in table 2.

***

---

## [Editor Report · Decision Letter 5]

29 Jun 2021

Dear Dr Lam, 

On behalf of my colleagues and the Academic Editor, Dr Bates, I am pleased to inform you that we have agreed to publish your manuscript "Long-term solid fuel use and risks of major eye diseases in China: a population-based cohort study of 486,532 adults" (PMEDICINE-D-20-04664R5) in PLOS Medicine.

There appears to be a small misunderstanding regarding use of the word "gender", which appears several times in your ms. PLOS Medicine's policy is to request use of the word "sex" in place of "gender" where it is appropriate to do so. Therefore, please adapt the language used on this point as appropriate prior to final acceptance. 

PRESS

Sincerely, 

Richard Turner, PhD 

rturner@plos.org